# Antibodies Targeting Human or Mouse VSIG4 Repolarize Tumor-Associated Macrophages Providing the Potential of Potent and Specific Clinical Anti-Tumor Response Induced across Multiple Cancer Types

**DOI:** 10.3390/ijms25116160

**Published:** 2024-06-03

**Authors:** Stephen Sazinsky, Mohammad Zafari, Boris Klebanov, Jessica Ritter, Phuong A. Nguyen, Ryan T. Phennicie, Joe Wahle, Kevin J. Kauffman, Maja Razlog, Denise Manfra, Igor Feldman, Tatiana Novobrantseva

**Affiliations:** 1Verseau Therapeutics, 2000 Commonwealth Ave., Auburndale, MA 02466, USA; 2Alloy Therapeutics, 275 Second Ave., Suite 200, Waltham, MA 02451, USA; 3Sanofi, 55 Corporate Driver, Bridgewater, NJ 08807, USA; 4HotSpot Therapeutics, One Design Center Pl. Suite 19-600, Boston, MA 02210, USA

**Keywords:** macrophage, immunotherapy, VSIG4

## Abstract

V-set immunoglobulin domain-containing 4 (VSIG4) is a B7 family protein with known roles as a C3 fragment complement receptor involved in pathogen clearance and a negative regulator of T cell activation by an undetermined mechanism. VSIG4 expression is specific for tumor-associated and select tissue-resident macrophages. Increased expression of VSIG4 has been associated with worse survival in multiple cancer indications. Based upon computational analysis of transcript data across thousands of tumor and normal tissue samples, we hypothesized that VSIG4 has an important role in promoting M2-like immune suppressive macrophages and that targeting VSIG4 could relieve VSIG4-mediated macrophage suppression by repolarizing tumor-associated macrophages (TAMs) to an inflammatory phenotype. We have also observed a cancer-specific pattern of VSIG4 isoform distribution, implying a change in the functional regulation in cancer. Through a series of in vitro, in vivo, and ex vivo assays we demonstrate that anti-VSIG4 antibodies repolarize M2 macrophages and induce an immune response culminating in T cell activation. Anti-VSIG4 antibodies induce pro-inflammatory cytokines in M-CSF plus IL-10-driven human monocyte-derived M2c macrophages. Across patient-derived tumor samples from multiple tumor types, anti-VSIG4 treatment resulted in the upregulation of cytokines associated with TAM repolarization and T cell activation and chemokines involved in immune cell recruitment. VSIG4 blockade is also efficacious in a syngeneic mouse model as monotherapy as it enhances efficacy in combination with anti-PD-1, and the effect is dependent on the systemic availability of CD8^+^ T cells. Thus, VSIG4 represents a promising new target capable of triggering an anti-cancer response via multiple key immune mechanisms.

## 1. Introduction

Under the influence of malignant cells, macrophages initially provide a niche for cancer development, ultimately helping to form and sustain both tumor growth and the associated tumor microenvironment (TME) [1]. Immune cells and mediators are universal components of TME, and TAMs serve as intermediaries between cancer and the immune system. In the early stages of tumorigenesis, classically activated macrophages can kill cancer cells and coordinate innate or adaptive immune cell-mediated mechanisms against the tumor. By contrast, in established tumors, macrophages have a predominantly tolerogenic or alternatively activated phenotype contributing to tumor survival, proliferation, neo-angiogenesis, metastasis formation, and active suppression of innate and adaptive immune responses [1]. TAMs function as immune suppressors by producing inhibitory mixtures of cytokines, presenting tumor-associated antigens in a tolerogenic fashion, directly or indirectly suppressing tumor-recognizing T cells via inhibitory molecules, producing chemokines such as CCL2 and CCL5 that preferentially recruit regulatory T (Treg) cells to the tumor site, and by activating metabolic pathways used predominantly by suppressive macrophages [1]. Given that TAMs support most tumor functions, it is compelling to look for strategies that can turn alternatively activated tumor-promoting macrophages into classically activated macrophages that can initiate an anti-tumor response.

A limited number of TAM-associated inhibitory molecules have been reported that can be found across many TAM phenotypes and in different cancer indications. We turned our attention to VSIG4, a molecule expressed exclusively by macrophages. Within TME, VSIG4 is specifically expressed in TAMs [2], and this expression has been shown to be associated with a poor prognosis in multiple tumor types, including non-small cell lung cancer, gastric cancer, multiple myeloma, ovarian cancer, and glioma, suggesting an important broad functional role at the tumor–immune interface [3,4,5].

VSIG4 (also known as CRIg or Z39Ig) is exclusively expressed on select tissue macrophages, acting as an inhibitory molecule suppressing the cells expressing it, mediating suppression of T cell proliferation and cytokine production and induction of organ-specific T and NKT cell tolerance [6,7]. VSIG4 is a member of the VSIG family of transmembrane proteins, structurally related to the B7 family [8]. The VSIG family has 8 members known to date, including VSIG1, VSIG2, VSIG3 (a ligand of VISTA), VSIG4, VSIG8, VSIG9 (also known as TIGIT), VSIG10, and VSIG10 L [8]. Human VSIG4 was first cloned from a human fetal cDNA library using degenerate primers recognizing Ig domains. Sequencing revealed an open reading frame of 400 amino acids with similarity to Z39Ig, a type 1 transmembrane protein [9]. The extracellular region of this molecule was found to consist of two Ig-like domains, comprising an N-terminal V-set domain and a C-terminal C2-set domain. Subsequently, a splice variant was identified, which lacks the membrane proximal IgC domain and is 50 amino acids shorter. VSIG4 binds to complement C3b and iC3b as well as C3b/iC3b-coated particles [10]. While the functionality of murine VSIG4 binding to complement C3 fragments in pathogen clearance has been demonstrated [10], the functionality of C3/VSIG4 interaction in the TME has yet to be revealed. We hypothesized that given its specific expression in TAMs, VSIG4 might have a role in the TME maintaining tolerogenic, M2-like macrophage phenotypes, and that inhibiting VSIG4 could lead to TAM and, consequently, TME reprogramming. It is unknown if the macrophage suppressive role of VSIG4 is dependent on C3 fragment binding.

In this study, we present data showing that inhibiting VSIG4 by a monoclonal antibody or by siRNA relieves VSIG4-mediated macrophage suppression and leads to repolarizing TAMs to an inflammatory phenotype capable of coordinating an anti-tumor immune response in preclinical settings. In in vitro and ex vivo assays, anti-hVSIG4 antibodies repolarize M2 and M2-like macrophages, respectively, and induce an immune response, culminating in T cell activation. More specifically, targeting VSIG4 upregulates pro-inflammatory cytokines and chemokines in primary human M2c macrophages. To assess the effects of VSIG4 inhibition in a relevant translational model, fresh patient-derived tumor samples were treated ex vivo with an anti-VSIG4 monoclonal antibody. Across multiple tumor types, anti-VSIG4 treatment resulted in a significant upregulation of cytokines involved in TAM repolarization and T cell activation and chemokines involved in immune cell recruitment. In vivo, anti-VSIG4 treatment inhibited tumor growth in a syngeneic mouse model either as a monotherapy or in combination with anti-PD-1 accompanied by macrophage repolarization and subsequent T cell activation.

Taken together, these data suggest that VSIG4 represents a promising new target capable of stimulating an anti-cancer response via multiple key immune mechanisms. We also describe an anti-hVSIG4 antibody that is well positioned to test the benefits of VSIG4 inhibition in clinical trials.

## 2. Results

### 2.1. VSIG4 Is Highly Expressed on Tumor-Associated Macrophages

By analyzing VSIG4 mRNA expression in a published single-cell RNA-seq (scRNA-seq) dataset comprising tumor-derived and peripheral immune populations from 7 non-small cell lung cancer (NSCLC) patients [11], we found that VSIG4 is localized to tumor-associated myeloid populations within the tumor microenvironment and is absent in every other cellular population in the TME and the periphery (Figure 1A). This finding prompted us to extend our analysis to approximately 10,000 human tumors with bulk mRNA profiling by RNA-seq spanning 33 cancer types that are captured in The Cancer Genome Atlas (TCGA) dataset [12]. In this dataset, we consistently found VSIG4 expression in a diverse array of human tumor types. The highest VSIG4-expressing tumor types included glioblastoma, mesothelioma, NSCLC, and pancreatic adenocarcinoma (Appendix A). These cancers are among the tumor types known for an abundant TAM infiltrate correlating with poor survival [13]. In all these tumor types, VSIG4 expression correlated with the expression of the inhibitory macrophage marker CD163 (Appendix A), supporting that VSIG4 is mostly expressed on suppressive macrophage populations within the tumor microenvironment.

To test the TCGA-predicted association of VSIG4 and CD163 mRNA expression, in situ hybridization (ISH) was performed on tumor microarrays from 10 tumor types to visualize VSIG4 mRNA and CD68/CD163 protein expression detected by IHC on TAMs (Appendix A). By quantifying VSIG4 fluorescence on 120 cores from 10 different tumor types, VSIG4 was found to be specifically expressed on a subset of CD163^+^ macrophages to varying degrees across most cores (75% or more) in lung, melanoma, kidney, head and neck, colorectal, ovarian, and pancreatic tumors and in a smaller fraction of cores in breast, bladder, and prostate tumors (Appendix A).

To further validate the association of VSIG4 and CD163 expression on the protein level, we stained ascites-derived macrophages for both markers and found that inhibitory TAMs expressing high levels of CD163 also expressed high levels of VSIG4 (Figure 1B). Additionally, we have differentiated primary human blood monocytes into M1, M0, and M2c macrophage subtypes (Figure 1C). We found that pro-inflammatory M1 macrophages do not express VSIG4; M-CSF-only differentiated slightly inhibitory M0 macrophages express low levels of VSIG4; while fully differentiated inhibitory M2c macrophages substantially upregulate VSIG4 (Figure 1D).

### 2.2. VSIG4 Isoforms Are Differentially Distributed between Normal Tissues and Cancer

Given the potential role of VSIG4 in human cancer, we compared *VSIG4* mRNA expression between normal tissues and cancer. Using the GTEx database for normal tissue [14] and TCGA for cancer [12], we observed no major differences in total VSIG4 mRNA expression levels on average between healthy and cancerous tissues. Only in several tissues, such as the brain, pancreas, and ovary, we see higher expression in cancer (Appendix A). Four *VSIG4* isoforms have been identified, one of which is an incomplete coding sequence (Appendix A). Two of the three complete isoforms have been well described in the literature and are often denoted as ‘long’ and ‘short’. The long isoform contains a variable (V-type) and a constant (C2-type) extracellular region while the short isoform only has the IgV extracellular domain. On the intracellular side, both isoforms contain a putative cAMP/cGMP-dependent protein kinase site and a protein kinase C phosphorylation site (S-311 in ‘long’/S-217 in ‘short’ and T-333 in ‘long’/T-239 in ‘short’, respectively). The third complete isoform, denoted as ‘long w/o C’ below, is similar to the ‘long’ one but lacks the T-333 intracellular protein kinase C phosphorylation site [15]. We looked at expression patterns of the three complete *VSIG4* isoforms at the transcriptional level. Comparing overall expression patterns between cancer and normal tissues, the ‘long’ isoform becomes more abundantly expressed in cancer while the truncated ‘long’ isoform, ‘long w/o C’, becomes less abundant (Appendix A). Even in tissues with a higher overall level of *VSIG4* mRNA in comparison to cancer, we observe that the ‘long’ isoform becomes higher expressed in cancer and the ‘long w/o C’ becomes less expressed. We demonstrate this in lung and breast tissues as examples (Figure 2). However, the switch from the ‘long w/o C’ to ‘long’ isoform in cancer is observed across multiple tissues that we were able to compare. The cancer-specific isoform of *VSIG4* that we have observed could have a functional significance in cancer and needs to be investigated further outside the scope of this study. At the same time, this isoform switch independently implies a functional role for VSIG4 itself in cancer.

### 2.3. Generation of Anti-VSIG4 mAb 12A12

Given the described negative role of VSIG4 in cancer, we set out to generate an inhibitory antibody for potential therapeutic applications. To achieve that, we started by generating a panel of VSIG4 antibodies that recognize both VSIG4 extracellular domain isoforms.

A diverse set of anti-VSIG4 Fabs was generated from mouse immunizations with recombinant VSIG4 protein and the phage display screening of the resulting antibody libraries. 12A12 was identified as a Fab that bound specifically to human VSIG4 expressed on the cell surface and to both human VSIG4 extracellular domain isoforms (‘long’ and ‘short’, see below) and cynomolgus monkey VSIG4 (Table 1). To confirm the binding properties of the 12A12 Fab identified from the phage display screening, a chimeric murine/human 12A12 antibody (12A12c) was recombinantly produced by transient HEK293 expression. 12A12c consists of the murine 12A12 variable regions and human IgG4 (S228P) heavy and kappa light chain sequences. A humanized version of 12A12c, termed 12A12h, was generated by CDR grafting of the 12A12 CDRs into human acceptor frameworks and backmutation of framework residues predicted to be important in CDR conformation following the approach previously described [16].

12A12c binds both the long and short isoforms of recombinant human VSIG4 (Figure 3A) as well as VSIG4 overexpressed on CHO cells (Figure 3B) and VSIG4-expressing monocyte-derived M2c macrophages (Figure 3C). Binding to both ‘long’ and ‘short’ VSIG4 isoforms demonstrates that 12A12 binds to the conserved IgV domain common to both isoforms. The affinity and kinetics of 12A12h and 12A12c binding to the recombinant human full-length VSIG4 extracellular domain protein was measured using biolayer interferometry (BLI) (Figure 3D, Table 1). 12A12c binds VSIG4 with a Kd of 19.4 nM, while 12A12c binds VSIG4 with a Kd of 7.4 nM, primarily driven by slower dissociation kinetics compared with the parental chimeric antibody. The ability of 12A12h to compete with C3b and iC3b ligands for binding to the human VSIG4 extracellular domain was evaluated in an ELISA format. 12A12h exhibits dose-dependent competition with both human C3b and human iC3b for binding to VSIG4 (Appendix A), in contrast to the anti-VSIG4 antibody 15A06 and isotype control. 15A06 binds the VSIG4 IgC2 domain, consistent with C3b binding to the VSIG4 IgV domain [17]. 12A12 has specificity for human and nonhuman primate VSIG4 but not for the related proteins. 12A12 does not bind the murine VSIG4 ortholog (Figure 3A), which shares 80% sequence identity within the conserved IgV domain, as well as to the B7-family related proteins, which share approximately 20% identity with the VSIG4 ECD (Appendix A). While structurally related to other VSIG family members, the VSIG4 ECD shares a similarly low sequence identity to VSIG family members as to B7-related proteins (high of 26% to VSIG8, Appendix A). 12A12h also does not have appreciable binding to molecules used to identify antibodies with nonspecific binding that correlates with poor in vivo PK, including baculovirus particles, calf thymus dsDNA, and human insulin (Appendix A).

### 2.4. VSIG4 Inhibition by siRNA or Blockade by the Antibody Results in Pro-Inflammatory Reprogramming of Suppressive Macrophages

As described above, VSIG4 expression correlates with the presence of CD163^+^ TAMs, associated with a suppressive, M2-like phenotype [18]. Both antibody blockade of VSIG4 and siRNA-mediated knockdown of the molecule were used to study the function of VSIG4 on human macrophage biology. Interestingly, 12A12c antibody treatment of inhibitory human macrophages phenocopied siRNA knockdown of VSIG4 as observed by the decreased expression of CD163 and CD16 on 12A12c-treated M2c macrophages after verification that VSIG4 was successfully downmodulated by treatment of immune-silent and potent siRNA to human VSIG4, encapsulated in lipid nanoparticles (Figure 4). The siRNA knockdown does not cause a large amplitude change, as it takes time for the reduced mRNA levels from the knockdown to translate into a decrease in the amount of VSIG4 protein. The importance of this data is to help interpret the 12A12c antibody effect directionally. This data support the idea that VSIG4 plays a modulatory role in maintaining the suppressive phenotype of M2-macrophages and that the 12A12 antibody is an antagonist of VSIG4. We then assessed whether the decreased M2-phenotype (CD163) correlated with a functional switch to a pro-inflammatory state. M2c macrophages were incubated with 12A12c and then stimulated with LPS to induce secretory activity, and soluble mediator levels were evaluated 24 h post-LPS treatment. 12A12c induced secretion of IL-6, IL-12, CCL3, CCL4, and CXCL10 across all 5 donors with the highest average induction of 6.5-fold, 2.7-fold, 3-fold, 2-fold, and 3.4-fold, respectively, between 0.37 and 1.1 μg/mL antibody concentrations with a consistent decrease in mediator induction at higher antibody concentrations (Figure 4D). Taken together, the data demonstrate that 12A12c induced the secretion of pro-inflammatory mediators from in vitro-derived M2c macrophages, indicating that 12A12c has repolarized M2 macrophages toward an M1 phenotype, even in the presence of M2c differentiating factors (M-CSF and IL-10), supporting the ability of anti-VSIG4 antibodies to override strong M2 differentiation and polarization signals. Thus, VSIG4 inhibition by two different modalities (siRNA and mAb) repolarizes M2c macrophages as measured by expression of M2 and M1 phenotype and inflammatory states.

### 2.5. VSIG4 Inhibition Leads to a Pro-Inflammatory Shift in Fresh Human Tumor Cultures

To evaluate the functional activity of 12A12c in the human tumor microenvironment, we utilized fresh human tumor cultures. Fresh human tumor cultures represent a relevant and unique approach to testing potential therapeutics for effects on human cancer [19,20,21]. Fresh tumor tissues obtained from surgical resection were dissociated and cultured in a dish in several wells per tumor sample, each well subjected to a different experimental or control treatment. These tumor cultures contain all cell populations present in the tumor microenvironment, such as tumor, immune, and stromal cells. Tumor cultures also preserve other factors present in the tumor microenvironment, such as tumor-specific mixtures of cytokines, chemokines, and growth factors, as well as multiple known and unknown ligands for the relevant immune-suppressive targets. Due to the pan-cancer expression of VSIG4 on tumor-associated macrophages, tumor tissue from all solid tumor types was accepted as it became available over the duration of the study.

The resultant dataset consists of tumors from twelve patients representing six different cancer types, which sample the continuum of what is currently considered immunogenic and nonimmunogenic tumor types: three lung tumor tissues, two breast tumor tissues, two kidney tumor tissues, two ovarian tumor tissues, two uterine tumor tissues, and one peritoneal tumor sample.

Principal component analysis (PCA) of secreted cytokine and chemokine levels from all samples showed that clustering occurred based on the individual tumor tissue (Appendix A). Tumor tissues from the same organ did not cluster together. This suggests that each individual tumor tissue had a unique composition of tumor, immune, and stromal cells, which exerted the greatest influence on the levels of cytokines/chemokines secreted in the culture, more so than the tissue of tumor origin. The mediators driving the major source of variation (the first principal component of the data or PC1) were CCL4, CCL3, TNFa, and IL-8 (Appendix A), which are all pro-inflammatory cytokines and chemokines mainly produced by myeloid immune cells. The second orthogonal source of variation (PC2) was driven by IL-2 (Appendix A), which is mainly produced by activated T cells. In this set of tumor cultures, the levels of secreted IFNγ were below the lower limit of quantification.

Next, using PCA, we examined the overall changes in mediator levels induced by 12A12c or the anti-PD-1 antibody pembrolizumab compared with the isotype-matched control antibody (Figure 5A,B, Appendix A). The greatest inductions were seen in the mediators driving PC1 of the fold change data: CCL3, TNFa, CCL4, and IL-6 (Figure 5C). It is worth noting that the mediators associated with PC1 of the absolute level and fold change data are very similar and related to myeloid biology. This means that the treatments are inducing major biological changes differentiating these samples and that this biology is related to myeloid cells. The greatest changes were induced by 12A12c rather than pembrolizumab. While it is not surprising that 12A12c induced the inflammatory myeloid-related cytokine program, it is the largest induction in an absolute sense when all treatment arms are considered, including that of pembrolizumab. It is possible that pembrolizumab has not produced a stronger change because this set of tumors happened to be immunologically ‘cold’, as evidenced by IFNγ levels below the lower limit of detection across all tumor samples. At the same time, turning ‘cold’ tumors ‘hot’ is the most important unmet clinical need of immuno-oncology now, the highest bar for TAM repolarizers, and the best promise of their clinical success.

12A12c showed activity across multiple tumors and tumor types (Figure 5). 12A12c induced a positive shift in the pro-inflammatory myeloid-associated PC1 signature (driven by CCL4, CCL3, and TNFa) (Appendix A). At 10 μg/mL 12A12c concentration, on average, the difference in PC1 scores between the treatment and control samples was comparable to the difference observed with 10 μg/mL pembrolizumab. At 0.1 μg/mL 12A12c concentration, 4 out of 11 (36%) tumors showed a response (difference in PC1 scores between the treatment and control samples) beyond experimental noise (Appendix A, red dashed line). This is more than 1 responder seen for 10 μg/mL 12A12c and 2 responders seen for 10 μg/mL pembrolizumab.

Similar patterns were seen for fold inductions of individual cytokines and chemokines comprising a myeloid-associated pro-inflammatory signature (Appendix A). On average, the greatest inductions were seen for CCL3, CCL4, and IL-6 at the lowest 12A12c concentration (0.1 μg/mL). For TNFa, induction levels were not dose-dependent at the three 12A12c concentrations studied. Induction of IL-2 showed a different dose-dependent pattern—the middle concentration of 12A12c (1 μg/mL) showed the greatest induction (Appendix A). The prevalence of response in this assay needs to be interpreted with caution given many of these tumor tissues are undergoing necrosis and may not be able to respond to any stimulation. Relative improved rate of response over pembrolizumab in the same cohort of tumors suggests a potential for clinical efficacy.

We examined the characteristics of the four most responsive tumors identified in Figure 5: two kidney tumors, one lung tumor, and one peritoneal tumor. All four tumors are among the seven tumors characterized by higher levels of myeloid infiltration estimated from baseline myeloid-associated PC1 scores across all samples (Appendix A, boxed). This suggests that in the ex vivo fresh dissociated tumor culture assay, a high level of basal myeloid infiltration is the requirement for demonstrating the functional activity of 12A12c. If the abundance of myeloid cells is too low, 12A12c effects cannot be observed above experimental noise in the format and 48 h timeframe of the assay. The observation is in line with the need for basal levels of T cell infiltration for the activity of T cell checkpoint inhibitors. This allows for formulating an initial biomarker hypothesis for an anti-VSIG4 therapy clinical development.

### 2.6. VSIG4 Blockade Is Efficacious in a Syngeneic Mouse Model as Monotherapy, Enhances Efficacy in Combination Therapy with Anti-PD-1, and Is Dependent on the Systemic Availability of CD8^+^ T Cells

To determine if suppressive macrophage repolarization to a pro-inflammatory phenotype translates into an in vivo effect, an anti-mVSIG4 antibody was evaluated in a mouse fibrosarcoma syngeneic mouse model, Sa1/N. Since 12A12 does not cross-react with the murine VSIG4 ortholog (Figure 3A), an anti-mouse VSIG4 antibody, 08A09, was generated via immunization of chickens with a recombinant mouse VSIG4 protein and subsequent phage display library generation and screening. The chimeric mIgG1 08A09 antibody binds specifically to mouse VSIG4 (Appendix A) with an affinity (Kd) of approximately 3.1 nM (Appendix A) and leads to repolarization of murine peritoneal macrophages from an M2-like to a more pro-inflammatory M1-like phenotype when administered in vivo (Appendix A). Therefore, 08A09 was chosen for assessing the efficacy of blocking VSIG4 in a mouse tumor model. 08A09 was evaluated for efficacy in the syngeneic mouse tumor model, Sa1/N [22,23], to determine if the anti-VSIG4-mediated repolarization of macrophages would lead to tumor growth reduction. Importantly, 08A09 was very well tolerated, did not lead to any adverse reactions in the model, and there was no reduction in body weight in any of the treatment groups. Treatment of Sa1/N-tumor-bearing mice with 08A09 led to a statistically significant inhibition of tumor growth both alone (Figure 6A,B; *p* = 0.004 in Study 2 and *p* = 0.017 in Study 1) and in combination with anti-PD-1 (Figure 6A; *p* < 0.0001) compared with the isotype control group. The combination treatment was significantly better at inhibiting tumor growth compared with either 08A09 or anti-PD-1 monotherapy (*p* < 0.0001 or *p* = 0.036, respectively; Figure 6A). Further, consistent with the reported role of VSIG4 in inhibiting T cell function [3,24,25], it was expected that systemic VSIG4 inhibition will have a T cell-dependent effect. Indeed, depletion of CD8^+^ T cells diminished the efficacy of anti-VSIG4 (Figure 6B).

Next, we examined the molecular and cellular biomarkers associated with reduced tumor volumes seen in the anti-VSIG4 and anti-PD-1 treatment groups, by analyzing tumor tissues and blood serum collected mid-study (17 days after the first dose by which time the tumor sizes were not dramatically different between the treatment groups). Treatment with 08A09 decreased the percentage of suppressive granulocytic MDSCs (CD11b^+^Ly6G^+^Ly6CloMHCII^−^) among the tumor-infiltrating immune cells defined as viable CD45^+^ cells (Figure 7B). At the same time, treatment with 08A09 led to increased T cell activation, measured as % Ly6C^+^ cells among CD3^+^ T cells (Figure 7C). The degree of modulation by 08A09 approached that of the anti-PD-1, even for the fraction of activated T cells (Figure 7A–C).

Importantly, we noticed a correlation between the fractions of immune cell populations and tumor size: as seen in Figure 7D,E, there was a direct correlation between the tumor volume and the percentage of suppressive gMDSCs and an inverse correlation with activated Ly6C^+^ T cells. Higher gMDSC fraction was associated with larger tumor volumes (Spearman *r* = 0.85; Figure 7D) and higher T cell activation was associated with smaller tumor volumes (Spearman *r* = −0.81; Figure 7E). This suggests that the balance of gMDSCs and activated T cells define an immune profile within the tumor microenvironment that directly correlates with the tumor’s growth capacity. Even within the same strain of mice, every mouse has a different adaptive immune system and, therefore, has different abilities to effectively fight the tumor. Thus, a given mouse from the control group can have better tumor growth control than a given mouse from a treatment group. However, in all animals, tumor growth largely depends on the TME immune composition of that individual animal regardless of the treatment, and anti-VSIG4 and anti-PD-1 treatments tend to shift the TME immune composition toward increased immune surveillance and decreased tumor growth.

In the circulating blood, 08A09 led to a pro-inflammatory serum cytokine profile. Pro-inflammatory mediators were induced to higher (GM-CSF, IL-12, IL-18, IL-1b, and IL-22) or similar (IFNγ and IL-5) levels with 08A09 compared with those seen with anti-PD-1 (Figure 7F).

Data from the Sa1/N syngeneic mouse model presented here support the hypotheses that (1) antibody-mediated modulation of VSIG4 can lead to repolarization of macrophages in vivo from an M2-like to a more M1-like phenotype, and (2) repolarization of macrophages leads to a pro-inflammatory shift in the immune profile within the tumor microenvironment, resulting in reduction in tumor growth.

## 3. Discussion

VSIG4 is a unique molecule due to its specificity for highly differentiated macrophages. Not only does it mark select tissue-resident macrophages, likely engaged in quenching uncoordinated pro-inflammatory microbursts and maintaining tissue homeostasis, it is highly upregulated on TAMs across tumor types—making it one of a handful of molecules specific to suppressive myeloid cells within TME. In addition to its expression profile, VSIG4 functionality in cancer has been suggested by multiple prognostic studies across multiple cancer types, and recently VSIG4 polymorphism has been linked to susceptibility and functional status in rheumatoid arthritis [26]. It has been noted that the T cell inhibitory functionality and complement binding are rarely combined in one molecule functionality [27]. Such a combination of functions makes VSIG4 even more unique. Mechanisms of T cell inhibition by VSIG4 are still under active investigation, and a recent paper has demonstrated that there is a very narrow window where VSIG4 is directly regulating the activity of CD8^+^ T cells and that it is happening in a TCR/CD3 signal strength-dependent fashion [24]. The interaction partner on the T cell surface has yet to be identified.

In this study, we describe an antibody targeting human VSIG4 that relieves VSIG4-mediated primary human macrophage suppression. 12A12 is a blocking antibody as it replicates the activity of siRNA targeting VSIG4 in primary human macrophages. 12A12 mediates repolarizing TAMs to an inflammatory phenotype and coordinates an anti-tumor immune response in preclinical testing: both in vivo as well as ex vivo. In vivo experiments demonstrate that systemic exposure to a VSIG4-inhibiting antibody induces repolarization in the tumor microenvironment, along with inducing strong tumor growth inhibition. Given the lower degree of homology between mouse and human VSIG4, syngeneic studies must rely on a surrogate antibody that recognizes mouse VSIG4. While allowing for model systemic inhibition, mouse studies have limited translational power in immuno-oncology, specifically in the field of myeloid biology, due to the ill-conserved biology of innate immune cells between species.

For the therapeutic candidate molecule, we concentrated on a molecule with the best functionality and comparable reactivity between human and nonhuman primate species to allow for safety assessment in preclinical testing. We turned to studying the effect of human VSIG4 inhibition in primary human tumors freshly derived from the patient surgery and delivered to the laboratory on wet ice. These tumor samples can be processed and treated in parallel with several different treatments to compare the functional consequences within the same tumor microenvironment. This is a powerfully translatable and predictive system as it maintains all the cell types and signals present in the original tissue next to the tumor cells themselves. Across multiple tumor types, anti-VSIG4 treatment resulted in a significant upregulation of cytokines involved in TAM repolarization and T cell activation and chemokines involved in immune cell recruitment. This activity seemed to be dependent on the initial presence of tumor-associated myeloid cells. This functional activity is consistent with reprogramming macrophages, as evidenced by activity patterns like those observed in the pure in vitro primary macrophage functional assays. This set of tumors was immunologically ‘cold’, as evidenced by the absence of detectable IFNg across all samples. In this set of tumors, the functional activity of anti-VSIG4 compared favorably with pembrolizumab, supporting our expectation that 12A12c can induce inflammatory responses in ‘cold’ tumors turning them ‘hot’, which is the largest unmet clinical need in oncology today. This study also preclinically validates our current preliminary biomarker hypothesis that response to anti-VSIG4 may be dependent on the initial TME infiltration with myeloid cells. The strength of the VSIG4 antagonistic antibody effect in the set of primary human tumors gives hope that this is the agent that can be broadly applied across tumor types and patient populations. It is important to say that VSIG4 inhibition in primary human tumors as monotherapy induces both repolarization and T cell activation, potentially positioning the above-described anti-VSIG4 antibody as a monotherapy agent activating both myeloid and lymphoid arms of the immune response via a single drug. As VSIG4 expression is documented in tissue macrophages and Langerhans cells, one might anticipate the untoward effects of blocking VSIG4 on organ inflammatory status. While we did not have a chance to perform extensive toxicology testing, the cage-side observations and body weight data in multiple mouse tumor models did not raise any concerns. No skin inflammation, weight loss, or any other observations have been recorded in animals treated with the anti-VSIG4 antibody. Additionally, strong expression on Kupffer cells is more specific to mouse biology and, therefore, unlikely to be a concern for human and nonhuman primate biology. In fact, liver and skin expression is among the lowest across healthy tissues in humans (Appendix A).

When selecting a therapeutic candidate antibody, we put a lot of emphasis on the specificity, human/nonhuman primate cross-reactivity, both splice isoforms coverage and functionality in the repolarization assay performed in primary human macrophages derived from multiple donors. It was important not to presume which part of the molecule the antibody needs to block but to select among many binders for optimal functionality. While complement, especially complement C3, has emerged as a functional component of the TME [28], we cannot be certain that we know that VSIG4 only interacts with C3b/iC3b in the TME. While 12A12 blocks the interaction of VSIG4 with C3 (Appendix A), interfering with part of C3 biology in the TME, we are open to the idea that 12A12 functions by interfering with the interaction between VSIG4 with other potentially unknown ligands.

Our serendipitous discovery that *VSIG4* has a cancer-preferred isoform is worth noting. It further positions VSIG4 as a very important molecule for cancer progression given that multiple tumor types select a particular isoform. It will be important to compare the signaling from modified and unmodified isoforms. If the functional differences are established following this observation, it will be important to attempt to target the modified isoform specifically to further improve the specificity of anti-cancer treatments. As the cancer-specific difference in VSIG4 isoform is intracellular, it, unfortunately, could not be targeted specifically by the antibody; therefore, it remained outside of the scope of our work. We believe the functionality of this specific isoform highly prevalent at the tumor site will be revealed by future work in the field.

In conclusion, VSIG4 represents a uniquely promising new target capable of stimulating an anti-cancer response via multiple key immune mechanisms. In this work we described an anti-VSIG4 antibody 12A12, later named VTX-1218 in its humanized IgG4 format, that is well positioned to test the benefits of VSIG4 inhibition in clinical trials.

## 4. Materials and Methods

### 4.1. In Vitro Generation of Monocyte-Derived Macrophages

Human PBMC samples or whole blood Leukopaks were purchased from Research Blood Components, LLC (Watertown, MA, USA). Research Blood Components follows the American Association of Blood Banks guidelines for drawing donors. The donor population consisted of healthy males and females between the ages of 18 and 65. All donors completed a uniform blood donor history questionnaire. An IRB-approved consent form was obtained from each donor giving permission to collect their blood and use or sell it at the Research Blood Components discretion for research purposes. Confidentiality and donor identification was assured. For more information, please see https://www.researchbloodcomponents.com/about-us (accessed on 30 May 2022).

Monocytes were enriched from human PBMCs using a monocyte negative selection kit (EasySep Human Monocyte Enrichment Kit without CD16 Depletion, Cat # 19058, Stemcell Technologies, Cambridge, MA, USA). Enriched monocytes were differentiated and polarized into M2c, M1, or M0-like macrophages. First, 1 mL of a 4 × 10^5^ cells/mL suspension of monocytes in Iscove’s Modified Dulbecco’s Medium (IMDM) containing 10% FBS was added per well to 24-well plates and incubated at 37 °C overnight (day 0).

For monocyte differentiation, 1 mL fresh assay media containing 50 ng/mL human macrophage colony-stimulating factor (M-CS, Cat #574804, Biolegend, San Diego, CA, USA) was added per well for M0 and M2c while 50 ng/mL human granulocyte-macrophage colony-stimulating factor (GM-CSF, Cat # 572902, Biolegend, San Diego, CA, USA) was added per well for M1-like macrophage differentiation. A half-media change was performed on day 4 by removing 500 μL of assay media and replacing it with 500 μL fresh assay media containing M-CSF or GM-CSF. On day 6 of the assay, differentiated M2 cells were polarized to M2c macrophages with 1 mL of fresh media containing 50 ng/mL M-CSF and 10 ng/mL of interleukin-10 (IL-10, Cat #574004, Biolegend, San Diego, CA, USA) while M1 cells wells were polarized with 100 ng/mL LPS (Cat #tlrl-eblps, Invivogen, San Diego, CA, USA) and 20 ng/mL IFNg (Cat #570204, Biolegend, San Diego, CA, USA). For macrophage phenotype analysis, the supernatants were collected, and the macrophages were harvested and subjected to flow cytometry.

### 4.2. Macrophage Functional Assay

On day 7, M2c macrophages were treated with 12A12c or isotype control diluted in a 12-point titration curve, with titration performed in duplicate. The highest antibody concentration was 30 μg/mL, which was serially diluted 1:3 to the lowest concentration of 0.0005 μg/mL. Cells were incubated with the antibody for 30 min at 37 °C, stimulated with 0.1 ng/mL LPS (Cat #tlrl-eblps, InvivoGen, San Diego, CA, USA) in assay media containing 50 ng/mL M-CSF and 10 ng/mL IL-10, and then incubated for 24 h at 37 °C. Supernatants were collected and analyzed using a cytokine 25-plex human Luminex panel (Cat #LHC0009M, Invitrogen, Carlsbad, CA, USA). Raw median fluorescence intensities (MFIs) from the Luminex assay were exported using the FlexMAP 3D software and further analyzed in the R statistical computing platform (https://www.r-project.org/, accessed on 30 May 2022).

### 4.3. Sa1N Fibrosarcoma Syngeneic Mouse Model

To establish the mouse model, female 7-week-old A/J mice (Jackson Laboratories) were injected subcutaneously with 8 × 10^6^ Sa1/N fibrosarcoma tumor cells (ATCC CRL-2544) in 200 µL (100 µL cells in Dulbecco’s Modified Eagle Medium (DMEM) and 100 µL 50% Matrigel for the tumor development. Mice that developed tumors of approximately 70 mm^3^ were randomized into 5 groups of 10 mice per group. 08A09 (6.25 mg/kg, Study 1 and 5 mg/kg, Study 2), anti-murine PD-1 (10 mg/kg; RMP1-14 clone), and isotype control (10 mg/kg) antibodies were administered via the intraperitoneal route twice weekly for 3 weeks. For study 2, anti-CD8 (10 mg/kg) treatment was administered starting on day 2 to randomization at approximately 50 mm^3^ tumor volume and then dosed with therapeutic antibodies or isotype thereafter. Mice were monitored twice weekly for tumor growth using calipers following the initiation of treatment. For Study 1, a mouse IgG1 isotype antibody was used as a negative control for 08A09 (clone MOPC-21) and a rat IgG2a isotype was used as a negative control for anti-PD-1 (clone 2A3). Since the two isotype control antibodies resulted in near-identical tumor growth profiles, for Study 2B, MOPC-21 was used as a negative control for both 08A09 and anti-PD-1 groups. For cellular immunophenotyping of tumor tissues and serum analysis from Study 2, three mice per group were euthanized and the tumors were harvested mid-study on day 17 and processed for immune profiling by flow cytometry at Charles River on the same day mice were euthanized and serum was analyzed by Luminex.

### 4.4. Flow Cytometry of Tumors from the Sa1N Syngeneic Tumor Model

Sa1N model: Tumors were harvested from the Sa1N-treated mice and prepared for flow cytometry (CR) analysis at Charles River. In brief, to prepare single-cell suspensions, tumor samples were washed in RPMI-1640, cut, and transferred into C-tubes containing 5 mL of an enzyme mix. Tumor samples were dissociated using a gentle MACS™ Dissociator (Miltenyi Biotec, Auburn, CA, USA) as instructed by the manufacturer (https://www.miltenyibiotec.com/upload/assets/IM0001973.PDF, accessed on 25 June 2022). Single-cell suspensions were filtered through a 70-micron filter, washed with RPMI1640, and centrifuged (350× *g*, 5 min, 4 °C). Subsequently, cells were washed with 1X DPBS, centrifuged (350× *g*, 5 min, 4 °C), resuspended in 1X DPBS, and counted. Red blood cells were lysed twice using ACK lysing buffer, neutralized with 1X DPBS, centrifuged (350× *g*, 5 min, 4 °C), and resuspended in 1X DPBS. For immunophenotyping, 2 × 10^6^ cells were transferred to a LW96-well plate (Cat# 96-DCCL-05, Curiox Biosystems, Woburn, MA, USA) and settled for 50 min on ice and then washed in 1X DPBS using laminar flow on a Curiox HT1000 platform (12 cycles, 70 μL). Cells were incubated for 20 min on ice in the dark with a Fixable Viability Dye and washed in Becton Dickinson (BD) Stain buffer using laminar flow on a Curiox HT1000 platform (12 cycles, 70 μL). Purified anti-mouse CD16/32 antibody (1 μL/sample) was used for Fc receptor blocking (Cat# 101302, BioLegend, San Diego, CA, USA). Cells were then washed in BD Stain buffer using laminar flow on a Curiox HT1000 platform (12 cycles, 70 μL) and stained for 40 min on ice in the dark with a 50 μL of an antibody cocktail in BD Stain buffer consisting of extracellular markers from the lymphoid (CD45, CD3, CD4, CD8, CD49b, NK1.1, Granzyme B, CD25, B220, FoxP3, Ki67, and Viability Dye) and myeloid panels (CD45, CD3, Ly6G, Ly6C, CD206, B220, HLA-DR (MHCII), CD163, CD11b, VSIG4, and Viability Dye). Stained cells were washed in the BD Stain buffer using laminar flow on a Curiox HT1000 platform (12 cycles, 70 μL), then fixed and permeabilized using the FoxP3/Transcription factor staining buffer set (Cat# 00 5523-00, eBiosciences, San Diego, CA, USA) according to the manufacturer’s instructions. Cells were washed in freshly prepared 1X Perm/Wash buffer using laminar flow on a Curiox HT1000 platform (12 cycles, 70 μL), then stained intracellularly for GrB, Ki67, FoxP3, and CD206, respectively, on ice for 50 min in the dark. Cells were then washed with 1X Perm/Wash buffer and BD Stain buffer using laminar flow on a Curiox HT1000 platform (each 12 cycles, 70 μL). Fixed cells were acquired on an Attune NxT flow cytometry system with 4 lasers.

### 4.5. Multivariate Analysis of Flow Cytometric Data

Acquisition files in the flow cytometry standard (FCS) format 3.1 were exported from the flow cytometer and analyzed in the R statistical computing environment (v4.0.3). Theoretical and technical details of the data analysis approaches described below are adopted from [29].

### 4.6. Preprocessing of Flow Cytometric Data

The flowCore (v2.3.1; 13) R package was used to import, compensate, and transform FCS files. For data transformation, a special class of biexponential functions known as the logicle transformation was used. Marginal events, debris, and doublets were removed using the ‘openCyto’ (v2.2.0; 14) and ‘flowDensity’ (v1.24.0; 15) R packages. Marginal events were defined using forward scatter area (FSC-A) and side scatter area (SSC-A) parameters. Debris was defined using FSC-A. Doublets were defined using FSC-A and FSC-W. CD45^+^ immune cells were gated based on the ‘CD45-A’ intensity for each sample, then pooled across all samples for downstream multivariate analysis.

### 4.7. Analysis of CD45^+^ Tumor-Infiltrating Immune Cells

Algorithms based on clustering, dimensionality reduction, and trajectory inference fully switch from the univariate/bivariate analysis to a multivariate approach. These tools consider the distribution of all markers simultaneously in the whole dataset, overcoming many manual gating limitations. Prior to dimensionality reduction, forward and side scatter parameters (FSC-A, FSC-H, FSC-W, SSC-A, SSC-H, SSC-W) were normalized using the z-score method. Dimensionality reduction using the UMAP (Uniform Manifold Approximation and Projection) algorithm [30,31] was performed on a 20-dimensional data matrix (3 FSC parameters, 3 SSC parameters, 14 fluorescent markers). The ‘umap’ (v 0.2.7.0) R package was used with the default settings. Manual gating was performed on dimensional-reduced 2D UMAP maps, and phenotypically complex gated cell populations were further subjected to UMAP projection and manual gating. For each unique cell population per sample, the mean fluorescence intensity of each marker was calculated as the arithmetic mean of the *logicle*-transformed fluorescence intensities.

### 4.8. Statistics

Statistical analyses were performed on tumor volumes at the termination of the study using GraphPad Prism (v9.1.0). Tumor volumes from different groups were compared using two-way ANOVA tests with multiple comparisons using Bonferroni correction.

### 4.9. siRNA Knockdown of VSIG4 in M2c Macrophages

A panel of 10 specific siRNA duplexes was generated (AxoLabs, Kulmbach, Germany). To test the efficacy of these siRNAs, dose–response curves were generated in vitro in Hepa1-6 cell line transfected with Dual-Glo Luciferase (Cat# E2920, Promega, Madison, WI, USA) construct containing human VSIG4 gene sequence as an untranslated region for one of the Luciferase genes. The IC50 for the best duplex used for primary cell transfection was determined to be <5 pM. The siRNA sequences for the best-performing duplex were sense: 5′-cuuucuAcGuGAcucuucudTsdT-3′ and anti-sense: 5′-AGAAGAGUcACGuAGAAAGdTsdT-3′) (dT = thymidine, s = phosphorothioate, lower case = 2′-OMe modification). siRNA-LNPs were manufactured by mixing an aqueous phase containing siRNA in 10 mM citrate buffer (pH 3) with an ethanolic solution containing ionizable lipid C12-200 [32] (AxoLabs, Kulmbach, Germany), 1,2-distearoyl-sn-glycero-3-phosphocholine (DSPC, Avanti Polar Lipids, Alabaster, AL, USA), cholesterol (Sigma. St. Louis, MO, USA), and 1,2-dimyristoyl-sn-glycero-3-phosphoethanolamine-N-[methoxy(polyethylene glycol)-2000] (C14 PEG 2000, Avanti Polar Lipids, Alabaster, AL, USA) at a 50:10:38.5:1.5 molar ratio and 9:1 total lipid:siRNA weight ratio. The aqueous and ethanol phases were mixed together at a 3:1 volume ratio in a microfluidic chip device using the NanoAssemblr (Precision Nanosystems, Vancouver, BC, Canada). The resultant siRNA-LNPs were dialyzed overnight in a 20,000 molecular weight cutoff cassette against 1x PBS at room temperature. On average, siRNA-LNPs had a median diameter of approximately 60–70 nm as measured by nanoparticle tracking analysis (ZetaView, ParticleMetrix, Ammersee, Germany). The siRNA encapsulation efficiency (approximately 80–90%) was determined using a modified Quant-iT Ribogreen assay (Invitrogen, Carlsbad, CA, USA) as previously described [33].

Purified human monocytes were isolated as described above and 400,000 cells were plated in IMDM 10% FBS in 24-well TC plates. After 24 h of adherence, the media were removed and replaced with fresh IMDM 10% FBS with either 50 ng/mL GM-CSF for the M1 conditions or 50 ng/mL M-CSF for the M2 conditions. The monocytes were transfected by adding the siRNA-LNPs directly to the wells at a final concentration of 25 nM. Two days later (day 4), a half-media change was performed by removing 250 μL of supernatant and adding 250 μL of fresh M1 or M2 media and siRNA LNPS. Two days after that (day 6), all supernatant was removed and 500 μL of M2c polarizing media (50 ng/mL M-CSF + 10 ng/mL IL-10) was added. Two days later (day 8), the cells were harvested for bDNA analysis to confirm target knockdown by a plate-based qualitative branched DNA assay (QuantiGene SinglePlex assay Kit, Thermo Fisher, Cambridge, MA, USA). Cells were also analyzed via flow cytometry for functional markers of M1/M2 phenotype including CD163, CD206, and CD16. Day 8 supernatants were assessed for multiple cytokines and chemokines by Luminex bead-based multiplex cytokine array.

### 4.10. Ex Vivo Patient-Derived Human Tumor Cultures

Fresh human tumor tissue originating from the discarded sample was sent on wet ice in DMEM within no more than 24 h after surgery for culture. Tissue samples were provided by the NCI Cooperative Human Tissue Network (CHTN, https://chtn.cancer.gov/, accessed on 13 June 2022), the National Disease Research Interchange (NDRI, https://ndriresource.org/, accessed on 13 June 2022) or BioIVT (https://bioivt.com/, accessed on 13 June 2022). Tissue samples were obtained using all the applicable operating policies and procedures that protect the subjects from whom specimens are obtained. These policies and procedures are consistent with current regulations and guidance for repositories from the Office of Human Research Protections (OHRP, DHHS).

Fresh tumor tissue (minimum 0.5 g) was placed in cold Dulbecco’s Modified Eagle’s Medium (DMEM) within 60 min of surgical resection and processed less than 24 h post-resection. Each tumor was placed in a 100 mm × 20 mm tissue culture-treated petri dish containing 20 mL of cold Hanks-balanced salt solution (HBSS). Surrounding fat, fibrous, and necrotic areas from tumor samples were removed prior to cutting the tumor pieces of 2–4 mm. Tumor pieces were transferred into disposable tissue grinder tubes with 2 mL of DMEM and ground by rotating the handle while applying adequate pressure. Dissociated tumors were then filtered with 40 um cell strainers into 50 mL conical centrifuge tubes, and the tubes were brought to 50 mL with ice-cold DMEM. All further steps were performed on ice. Samples containing dissociated cells were centrifuged for 5 min at 300× *g* and the supernatant was discarded. Cells were washed twice with ice-cold DMEM by repeating the centrifugation step. After the last wash, cell pellets were resuspended in 1–5 mL ice-cold culture medium DMEM containing 8% FBS, 2% human serum, 100 IU/mL penicillin/streptomycin, 1 mM Glutamax, 55 uM 2-mercaptoethanol, 1X nonessential amino acids, 1 mM sodium pyruvate, 48 ng/mL human recombinant IL-2 (100 IU/mL, assuming a specific activity of 2.1 × 10^6^ IU/mg), and 1X insulin/transferrin/selenium (41-400-045, Fisher Scientific, Cambridge, MA, USA).

Each tumor suspension was plated at a cell density of 5 × 10^5^ cells/mL (2.5 × 5 cells/well in 500 μL/well) in 24-well plates and treated with 12A12c or Isotype control at 10, 1, or 0.1 μg/mL or pembrolizumab at 10 μg/mL final concentration and incubated at 37 °C/5% CO_2_ for 48 h. Supernatants were then collected for mediator analysis using a cytokine 25-plex human Luminex panel (Invitrogen Cat. # LHC0009M, Lot # 287587-008) with technical duplicates. Raw median fluorescence intensities (MFIs) from the Luminex assay were exported using the FlexMAP 3D software and further analyzed in the R statistical computing platform (https://www.r-project.org/, accessed on 30 May 2022). After the removal of outlying measurements from the calibration standards, standard curves were generated by fitting MFIs from standards to a 5-parameter log-logistic (5-PLL) function using nonlinear regression. Whenever the 5-PLL model did not converge, a 4-parameter log-logistic (4-PLL) model was used instead. Fitted analyte concentrations below LLOQ or above ULOQ were replaced with the LLOQ or ULOQ values, respectively. All fitted analyte concentrations were reported in pg/mL. Data QC and outlier detection were performed using principal component analysis (PCA). Thirteen out of twenty-five analytes in the Luminex panel were excluded from analysis because they were below the lower limit of quantification (LLOQ) in at least twelve out of thirteen tumors. Analyte concentrations from technical replicates were averaged for each tumor, and another PCA was performed on the resultant dataset using 10 out of the 25 analytes that were detected above the LLOQ and had a variance above 0.01 across treatment conditions. Change in PC1 score was calculated using the following formula: ΔPC1 = PC1 (mAb treatment) − PC1 (Isotype). Subsequently, fold changes for each tumor were calculated between 12A12c or pembrolizumab-treated samples and isotype control samples at matching antibody concentrations, and PCA was performed on the resultant fold change matrix. For each analyte, tumors were further excluded when the analyte levels across all conditions were below the lower limit or above the upper limit of quantification. For graphing and comparisons between different treatment groups, data from one tumor were excluded due to missing the 10 μg/mL isotype condition, which is the reference for both the 10 μg/mL 12A12c and 10 μg/mL pembrolizumab treatment groups. Graphing was performed in either R or GraphPad Prism software (https://www.graphpad.com/, accessed on 13 June 2022).

### 4.11. Antibody Generation

Murine anti-human VSIG4 antibodies were discovered by the phage display screening of murine Fab libraries derived from mice immunized with human VSIG4 (FairJourney Biologics, Porto, Portugal). Three BALB/c mice were immunized with His-tagged recombinant human long isoform VSIG4 protein (human VSIG4-L-His; derived from amino acids 20–283 of reference sequence Q9Y279) and boosted with recombinant cynomolgus monkey VSIG4 (cyno VSIG4-L-His; derived from amino acids 25 to 288 of reference sequence XP_005593850.1). Fab libraries were generated from spleen and lymph node RNA harvest, and total RNA was used as a template for cDNA synthesis by RT-PCR using the SuperScript^®^ III First Strand Synthesis System random priming (Invitrogen, Carlsbad, CA, USA). The cDNA products were used directly for polymerase chain reaction (PCR) amplification of the variable domain of the heavy chain (CH), the first constant domain of IgG heavy chain and a portion of the hinge (CH1-Hinge), the variable domain (Vκ), and the constant domain (Cκ) of the antibody kappa light chain using the Expand™ High Fidelity System (Roche, Basel, Switzerland). Per mouse, the genes codifying for the VH, CH1, and a portion of the hinge or the genes codifying for Vκ and Cκ were cloned separately into a phagemid leading to heavy chain and light chain sub-libraries, respectively. The complete Fab library was constructed by cloning the VHCH1-Hinge inserts obtained by endonuclease restriction digesting the heavy chain sub-library DNA into the phagemid vector containing the light chain sub-library and by electroporating electrocompetent *E. coli* TG1 cells. In total, three Fab display libraries, corresponding to each mouse immunized, were generated.

### 4.12. Phage Display Screening

A total of three consecutive rounds of phage display selections were performed to enrich for Fabs specific for (i) human VSIG4 or (ii) human and cynomolgus monkey VSIG4. A first round of phage display selections with 100 nM biotinylated human VSIG4 proteins captured on streptavidin magnetic beads was conducted, with selections conducted in parallel against human Fc1 and His tagged versions of both the long and short VSIG4 isoforms. A second round followed, using biotinylated VSIG4 proteins at 20 nM or 2 nM captured on streptavidin magnetic beads, with selections conducted in parallel against human and cynomolgus monkey VSIG4. A final round of panning was performed using CHO-K1 cells expressing the human long isoform VSIG4. All phage display selections were performed with total elution of the VSIG4 binding phage with trypsin. Selected antibodies enriched from phage display screening were expressed as mouse/human chimeras with the mouse variable regions and human IgG4 backbone containing an S228P heavy chain mutation paired with a human kappa light chain.

### 4.13. Affinity Measurements

Binding affinity was determined by biolayer interferometry (BLI), using a ForteBio Octet^®^ Red384 system. All samples were prepared in black 96-well flat bottom plates (Greiner, Cat. #655209) and diluted in the kinetics buffer (1x DPBS (Gibco, Cat. #LS14190250) containing 0.1% bovine serum albumin and 0.02% Tween^®^20). Anti-VSIG4 antibodies (ligand) were diluted to a final concentration of 10 μg/mL and captured on anti-human IgG Fc capture (AHC) biosensors (Cat. #18-5060, ForteBio, Fremont, CA, USA). Recombinant His-tagged human VSIG4 (h-VSIG4-L-His) was diluted to 100 nM. Biosensors were rehydrated prior to the assay in the kinetics buffer for 10 min and then equilibrated in the kinetics buffer for 60 s. The antibody was then immobilized for 300 s followed by a 120 s baseline wash in a fresh kinetics buffer, 300 s analyte association, and 300 s dissociation in the same buffer wells as the baseline wash step. Biosensors were regenerated over 5 cycles of 5 s each in the regeneration buffer (10 mM glycine, pH 2.0; Cat #BR100255, GE Healthcare Life Sciences, Marlborough, MA, USA) followed by neutralization in the kinetics buffer. All assay steps occurred at 30 °C with an acquisition rate of 5.0 Hz. Fitting was calculated using ForteBio Data Analysis HT software (https://www.sartorius.com/en/products/biolayer-interferometry/octet-systems-software, accessed on 29 April 2022), using a 1:1 global binding model for both association and dissociation. Processing parameters included reference subtraction with isotype control human IgG4 used for reference sensor subtraction.

### 4.14. Antibody Binding to VSIG4 Isoforms and Orthologs

Antibodies were characterized for binding to recombinant VSIG4 proteins by ELISA. MaxiSorp™ high protein-binding capacity 96-well microplates (Sigma Aldrich, St. Louis, MO, USA) were coated with human (long and short isoforms), cynomolgus monkey, and mouse VSIG4 proteins [human VSIG4-L-Fc, human VSIG4-S-Fc, cyno VSIG4-L-His, mouse VSIG4-Fc] at 1 µg/mL diluted in PBS. Coated microplates were blocked with dried skimmed milk at 4% in PBS. Coated proteins were then incubated with each purified antibody at a concentration of 50 nM for 1 h to allow for the binding to the proteins. Binding detection was measured using a goat anti-huIgG monoclonal detection antibody, HRP conjugated (Jackson Immune Research, West Gove, PA, USA). As a control for specificity, an irrelevant antibody and protein were used. OD at 450 nm was determined using a standard absorbance microplate reader.

### 4.15. C3b Ligand Competition Assay

Recombinant full-length VSIG4 extracellular domain protein (h-VSIG4-L-His) was immobilized on microtiter plates (Costar, Cat #3690, Thermo Fisher, Cambridge, MA, USA) at a concentration of 2 mg/mL overnight at 4 °C. Plates were then washed 3 times with 1x PBS with 0.05% Tween 20 (PBST, Cat #28352, Thermo Fisher, Cambridge, MA, USA), blocked with 3% bovine srum albumin (BSA, Cat #126593, Millipore Sigma, Temecula, CA, USA) in PBS for 1 h at 37 °C, and washed 3 times. Anti-VSIG4 and isotype control antibodies were titrated 3-fold from 81 nM to 0.0014 nM in 3% BSA and 0.05% Tween 20, PBS, pH 7.4 to generate 11-point dose–response curves for each antibody. Antibodies were then added to VSIG4-coated plates. Wells containing no antibodies were included as complete C3b/iC3b binding (zero competition) controls. Plates were incubated for 45 min at 37 °C followed by washing 3 times with PBST. Biotinylated C3b and iC3b were then directly added at approximate EC50 values (15 nM and 4 nM, respectively), diluted in 3% BSA and 0.05% Tween 20, PBS, pH 7.4. Wells containing no C3b or iC3b were included as background controls. Plates were incubated for 45 min at 37 °C followed by washing 3 times with PBST. Bound biotinylated C3b or iC3b was detected with streptavidin-HRP conjugate (Cat #016-030-084, Jackson ImmunoResearch, West Grove, PA, USA), and plates were developed with 3,3′,5,5′-Tetramethylbenzidine (TMB) substrate (Cat #5120-0077, SeraCare, Milford, MA, USA), quenched with 1 N hybrochloric acid (Cat #A4810212 HCL, Thermo Fisher, Cambridge, MA, USA), and OD450 values were read on a BioTek Cytation 5 instrument at an absorbance of 450 nm.

### 4.16. Binding to VSIG4 Overexpressing Cells

Anti-VSIG4 antibodies were tested for binding to wild-type (WT) CHO-K1 cells and to human VSIG4-expressing CHO-K1 cells for the determination of EC50 values. Three-fold serial dilution of each antibody starting at 200 nM was prepared in an assay buffer (0.5% of heat-inactivated fetal bovine serum with 1:1000 Ethylenediaminetetraacetic acid in PBS). Diluted samples were incubated with 1.5 × 10^5^ WT or human VSIG4-L expressing CHO-K1 cells for 1 h, and the bound antibody was detected using a goat anti-human IgG (Fc specific), FITC-conjugated polyclonal detection antibody (Sigma Aldrich, St. Louis, MO, USA). Samples were measured on an Attune™ NxT (Thermo Fisher Scientific, Waltham, MA, USA).

### 4.17. Anti-Mouse VSIG4 Antibody Derivation

Antibodies directed against murine VSIG4 were generated by similar methods as described above. Briefly, two chickens were immunized with His-tagged recombinant mouse VSIG4 protein (mouse VSIG4-His, derived from NP_808457.1 (amino acids 20–187 amino acids), and phage display Fab libraries were constructed and screened for binding to recombinant murine VSIG4 protein and 293T cells overexpressing mouse VSIG4. Selected antibodies enriched from the phage display screening were expressed as chicken/murine chimeras with the chicken variable regions and mouse IgG1 backbone. The binding affinity of chicken/murine chimeras was determined by biolayer interferometry (BLI), using a ForteBio Octet^®^ Red384 system. All samples were prepared in black 96-well flat bottom plates (Greiner, Cat. #655209) and diluted in the kinetics buffer (1x DPBS (Cat #LS14190250, Thermo Fisher, Cambridge, MA, USA) containing 0.1% bovine serum albumin, and 0.02% Tween^®^20). Anti-mouse VSIG4 antibodies (ligand) were diluted to a final concentration of 10 μg/mL and captured on anti-mouse IgG Fc capture (AMC) biosensors (Cat. #18-5088, ForteBio, Fremont, CA, USA). Recombinant mouse VSIG4 (m-VSIG4-His) was diluted to 100 nM. The sequence of m-VSIG4-His was derived from the reference sequence NP_808457.1 (amino acids 20–187). Biosensors were rehydrated prior to the assay in the kinetics buffer for 10 min and then equilibrated in the kinetics buffer for 60 s. The antibody was then immobilized for 300 s followed by a 120 s baseline wash in a fresh kinetics buffer, 300 s analyte association, and 300 s dissociation in the same buffer wells as the baseline wash step. Biosensors were regenerated over 5 cycles of 5 s each in the regeneration buffer (10 mM glycine, pH 2.0; Cat #BR100255, GE Healthcare Life Sciences, Marlborough, MA, USA) followed by neutralization in the kinetics buffer. All assay steps occurred at 30 °C with an acquisition rate of 5.0 Hz. Data fitting was performed using ForteBio Data Analysis HT software (https://www.sartorius.com/en/products/biolayer-interferometry/octet-systems-software, accessed on 29 April 2022). Processing parameters included reference subtraction, with isotype control mouse IgG1 (clone MOPC21) used for reference sensor subtraction. Y-axis data were aligned to the average baseline step for the last 5 s and all steps were inter-step corrected to the start of the dissociation step. High-frequency noise was removed using Savitzky–Golay filtering and the final fitting was a 1:1 global binding model to both association and dissociation.

To test for functional activity, we assessed whether select antibodies are able to repolarize mouse peritoneal macrophages in vivo. In brief, on day 1, 8-week-old female BALB/c (000651, JAX Laboratories) mice were dosed IP with 10 mg/kg of the 08A09, a known pro-inflammatory antibody against CD40 [34,35] or an isotype control antibody. Animals were euthanized on day 3 by CO_2_ inhalation and the peritoneal lavage was harvested in RPMI-1640 with 2% FBS to be analyzed by flow cytometry. Ice-cold PBS was added to the lavage samples where the volume was below 8 mL for a final volume of 9 mL. The samples were centrifuged 500× *g* for 5 min, and 100 μL of supernatants was placed in a 96-well plate and stored at −80 °C until analyzed using a Luminex 26-plex mouse assay (EPX260-26088-901). The pelleted peritoneal exudate cells (PECs) from the lavages were washed twice with 5 mL ice-cold PBS and resuspended in 1 mL cold RPMI for cell counting. Final volumes were adjusted for a cell concentration of 1 × 10^6^ cells/mL, and 100 μL per well was added to a 96-well plate for flow cytometry staining using a mixture of antibodies including CD163-FITC (Cat# 11-1631-82, eBioscience, San Diego, CA, USA), MHCII-PE/594 (Cat# 107648, Biolegend, San Diego, CA, USA), CD36-PE/Cy7 (Cat# 102616, Biolegend, San Diego, CA, USA), CD80-BV605 (Cat#104729, Biolegend, San Diego, CA, USA), CD4-BV650 (Cat# 100469, Biolegend, San Diego, CA, USA), and a BV510 Viability Dye (Cat# 65086614, Invitrogen, Carlsbad, CA, USA). A total of 100 μL (1 × 10^5^ cells) was added per well in a 96-well plate and PECs were cultured for 24 h in RPMI with 10% FBS. Supernatants were collected and analyzed for secreted mediators by Luminex.

## Figures and Tables

**Figure 1 ijms-25-06160-f001:**
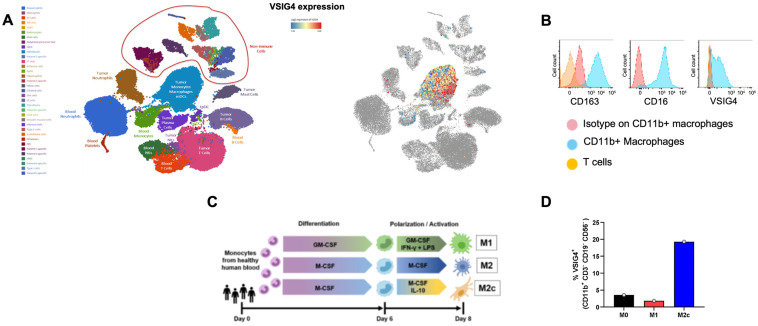
VSIG4 expression is restricted to tumor-associated myeloid populations. (**A**) Single-cell transcriptome atlas showing VSIG4 expression on human immune cells in non-small cell lung cancer. Published data [11] were downloaded and analyzed using the BioTuring Browser software (https://bioturing.com/, accessed on 11 March 2020). UMAP plot of cells from patient tumor biopsies (*n* = 7). (**B**) VSIG4 expression on macrophages in ascites fluid. Data are representative of 2 samples and are presented as the MFI of VSIG4 on CD11b^+^ macrophages. (**C**) Schematic of M1, M0, and M2c macrophage differentiation and polarization conditions using human peripheral blood monocytes. (**D**) VSIG4 expression on M0, M2c and M1-like macrophages. Data are representative of greater than 15 experiments and are presented as the percent of VSIG4 positive cells within CD11b^+^ lineage (CD3^−^CD19^−^CD56^−^) macrophages within each macrophage subset.

**Figure 2 ijms-25-06160-f002:**
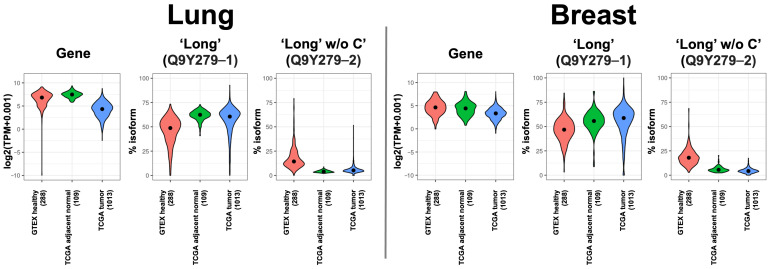
Relative abundance of *VSIG4* isoforms in healthy versus cancer lung and breast tissues. A switch from the ‘long w/o C’ isoform in healthy tissues (measured in the GTEX dataset, data in red) to the ‘long’ isoform in cancer (measured in TCGA dataset, adjacent normal tissue in green and tumor tissue in blue) is observed across most tissues that we were able to compare (see also Appendix A). Lung and breast cancers are presented as examples here with very low levels of ‘long w/o C’ and higher levels of the ‘long’ isoform in cancer.

**Figure 3 ijms-25-06160-f003:**
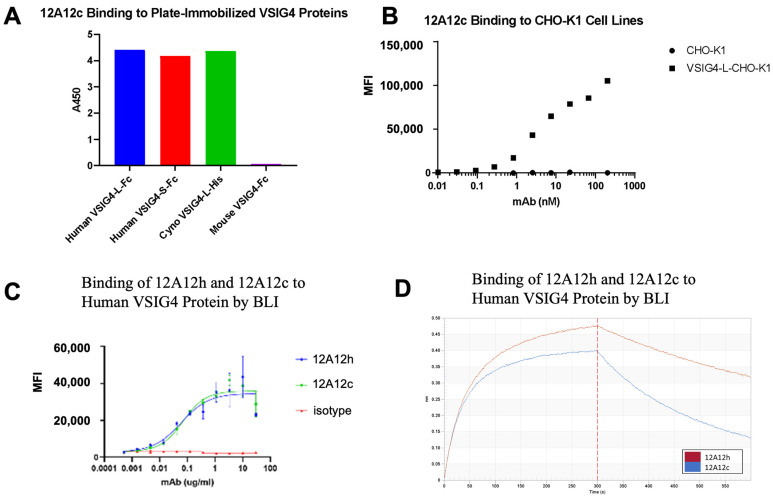
12A12 antibody characterization. (**A**) Binding of 12A12c to recombinant VSIG4 proteins. Recombinant human long (L) and short (S) isoform VSIG4, cynomologous monkey VSIG4, and mouse VSIG4 were immobilized on microtiter plates, and 12A12c binding was measured by ELISA. (**B**) Binding of 12A12c to VSIG4 expressed on cells. Parental CHO-K1 cells and CHO-K1 cells stably expressing VSIG4 were incubated with varying concentrations of 12A12c and the extent of binding measured by flow cytometry. The binding signal is measured as the median fluorescence intensity (MFI) of the cell population. (**C**) Binding of 12A12h and 12A12c to monocyte-derived M2c macrophages. In vitro-derived M2c macrophages were incubated with varying concentrations of the indicated antibody and the extend of binding was measured by flow cytometry. The binding signal is measured as MFI for CD11b^+^ CD163hi cells (M2) cells. (**D**) Binding affinity of 12A12h and 12A12c. 12A12h, 12A12c, and isotype control were immobilized onto anti-human Fc (AHC) capture sensors and evaluated for binding to 100 nM recombinant His-tagged VSIG4. Data shown are reference-subtracted sensorgrams for the association phase (0–300 s) and the dissociation phase (300–600 s) of His-tagged VSIG4 extracellular domain protein binding to 12A12h (red) and 12A12c (blue). The binding signal is measured as a nanometer (nm) shift.

**Figure 4 ijms-25-06160-f004:**
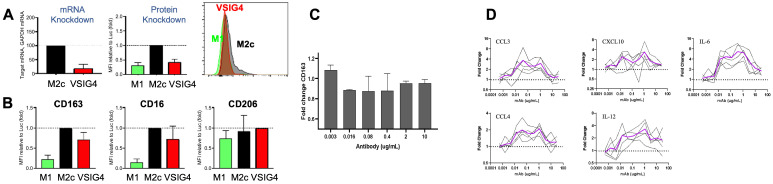
siRNA knockdown of VSIG4 attenuates the M2c phenotype, and 12A12c repolarizes M2c-macrophages toward an M1-like phenotype (**A**) siRNA knockdown of VSIG4 in M2c macrophages transfected with 25 nM VSIG4 siRNA LNPs on days 1 and 3 during the differentiation and polarization period. VSIG4 mRNA and protein levels were measured on day 8 by branched chain DNA analysis and flow cytometry, respectively (*n* = 4 donors). Green represents M1 macrophages, black represents M2 macrophages, red represents M2 macrophages transfected with VSIG4 siRNA. (**B**) The fold change of M2c markers CD16, CD163, and CD206 as determined by flow cytometry in M2c macrophages treated with siRNA LNPs relative to those treated with luciferase siRNA. (**C**) Flow cytometry determined the fold change of the M2c marker CD163 based on the % CD163^+^ of CD11b^+^CD3^−^CD45^+^ M2c macrophages treated with 12A12 for 24 h followed by 0.1 ng/mL LPS for 24 h. (**D**) The fold change based on pg/mL secreted proteins measured by a 25-plex Luminex assay from human monocyte-derived M2c macrophages treated with 12A12c or isotype control for 30 min at 37 °C followed by 0.1 ng/mL LPS for 24 h. Fold changes were calculated with respect to the human IgG4 isotype control antibody for each donor and each concentration. Black lines represent individual donors (*n* = 5). Purple lines represent the mean fold change across all donors.

**Figure 5 ijms-25-06160-f005:**
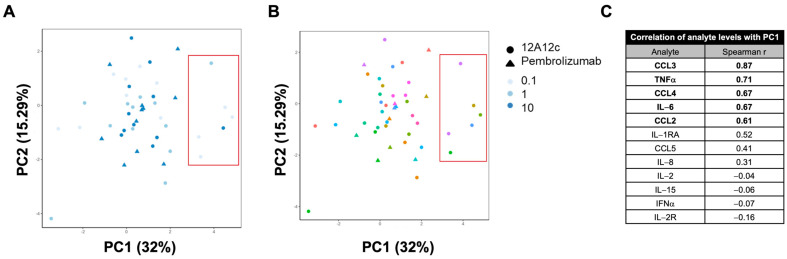
Principal component analysis (PCA) of fold inductions of secreted mediators in response to 12A12c or pembrolizumab. (**A**,**B**) PCA plots of log-transformed fold inductions compared with isotype control of secreted mediator levels in tumor culture supernatants analyzed by the Luminex assay from *n* = 12 tumors. Marker shapes represent different antibody treatments: 12A12c (circle), pembrolizumab (triangle). Marker colors indicate either antibody concentration with pale color representing 0.1 μg/mL, middle intensity 1 μg/mL and highest intensity 10 μg/mL (**A**) or the tumor tissue ID—each color representing one original tumor split into different treatments (**B**). Only 10 out of the 25 analytes that were detected above the LLOQ were included in the analysis. (**C**) Spearman correlation coefficients between PC1 score and log-transformed fold inductions in analytes. Analytes with absolute correlation coefficients higher than 0.6 are highlighted in bold.

**Figure 6 ijms-25-06160-f006:**
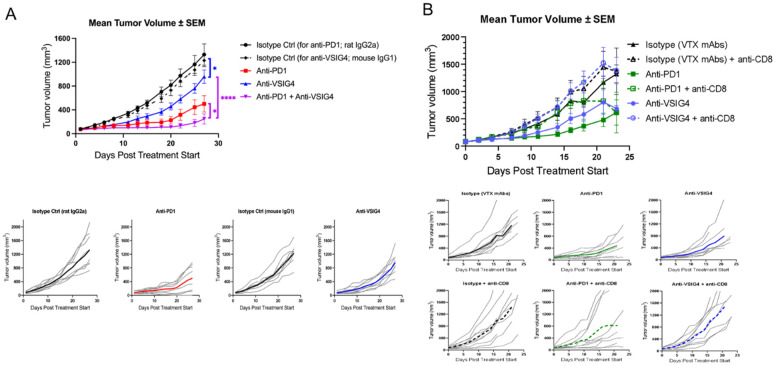
Efficacy of anti-VSIG4 in the Sa1N fibrosarcoma mouse model. Anti-mouse VSIG4 (08A09) was dosed at 5 mg/kg for Study 1 (**A**) and 6.25 mg/kg for Study 2. Different colors represent different treatments (black for isotype controls matching anti-PD-1 or anti-VSIG4 antibodies, blue anti-VSIG4 antibody, red anti-PD-1 antibody and purple the combination of both active treatments) and stars (* with *p* < 0.05, **** with *p* < 0.001) represent levels of significance for the differences between treatment groups. The data is shown as averages only (upper panels) and individual animal tumor volume lines with averages in color (bottom panels). (**B**) while anti-PD-1 and isotype control antibodies were dosed at 10 mpk 2x weekly for 27 (Study 2) or 23 (Study 1) days. Anti-CD8 was dosed when tumors reached ~50 mm^3^, 2 days prior to mouse randomization and other antibody treatments and then dosed with the corresponding antibodies as indicated in the figure. Data are presented as the mean tumor volume ± standard error of the mean (SEM) for all dose groups. Study 1: N = 10 mice per group through D17 and then N = 7 mice per group through D27. Study 2: N = 10 mice per group through D11 and then N = 8 mice per group through D23. Changes in group numbers reflect that mice were removed from the study mid-study for immunophenotyping analysis. Dosing was initiated on day 0 at an average tumor volume of ~70 mm^3^. Statistical analyses were performed using GraphPad Prism. Tumor volumes from different groups measured on day 21 (Study 1) or Day 27 (Study 2) were compared using two-way ANOVA tests with multiple comparisons using Bonferroni correction.

**Figure 7 ijms-25-06160-f007:**
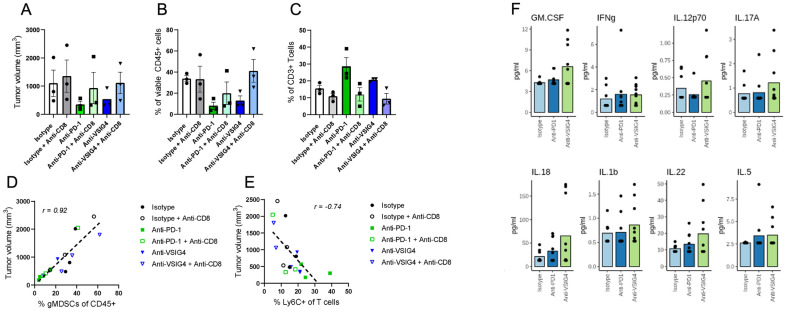
Anti-VSIG4 treatment modulates tumor cellularity and peripheral mediators. (**A**) Tumor volume (**B**) granulocytic MDSCs (CD11b^+^Ly6G^+^Ly6CloMHCII^−^) quantified as the percentage of total viable immune cells (CD45^+^) and (**C**) activated T cells quantified as the percentage of Ly6C^+^ cells among viable T cells (CD3^+^). (**A**–**C**) Bars represent the mean from *n* = 3 mice per group on day 17, and error bars represent standard deviations. (**D**,**E**) Tumor volumes across all treatment groups were correlated with gMDSCs (**D**) or T cell activation (**E**). Correlation coefficients were calculated using the Spearman method. Calculations for each mouse were then averaged and presented as average ± standard deviation. *n* = 3 mice per group. (**F**) Secreted serum cytokines and chemokines on day 17 post-treatment were analyzed by Luminex. Bars represent the mean from *n* = 10 mice per group.

**Table 1 ijms-25-06160-t001:** 12A12 chimera and humanized antibody affinity parameters. Binding kinetics of 12A12h and 12A12c measured by BLI. 12A12h, 12A12c, and isotype control were immobilized onto anti-human Fc (AHC) capture sensors and evaluated for binding to 100 nM recombinant His-tagged VSIG4 from the relevant species. Data shown are reference-subtracted sensorgrams fit to a 1:1 binding model.

Affinity parameters for 12A12h and 12A12c binding to recombinant human VSIG4.		
	12A12h	12A12c
Kd	7.4 nM	19.4 nM
On Rate (M^−1^s^−1^)	1.70 × 10^5^	1.95 × 10^5^
Off Rate (s^−1^)	1.25 × 10^−^^3^	3.79 × 10^−^^3^
R^2^	0.989	0.995
Affinity parameters for 12A12h binding to recombinant cynomolgus and rhesus VSIG4.		
	Cyno	Rhesus
Kd	8.1 nM	9.6 nM
On Rate (M^−1^s^−1^)	1.61 × 10^5^	1.74 × 10^5^
Off Rate (s^−1^)	1.30 × 10^−^^3^	1.67 × 10^−^^3^
R^2^	0.991	0.990

## Data Availability

Data is contained within the article and Appendix A.

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
