# Peer review of "Antibodies Targeting Human or Mouse VSIG4 Repolarize Tumor-Associated Macrophages Providing the Potential of Potent and Specific Clinical Anti-Tumor Response Induced across Multiple Cancer Types"

_ijms, 2024, doi:10.3390/ijms25116160_

Round 1

Reviewer 1 Report (Previous Reviewer 2)

Comments and Suggestions for Authors

The manuscript “Antibodies Targeting Human or Mouse VSIG4 Repolarize Tumor-associated Macrophages Providing the Potential of Potent and Specific Clinical Anti-tumor Response Induced Across Multiple Cancer Types” investigates the role of the V-set and immunoglobulin domain-containing 4 (VSIG4) protein, a B7 family-related protein exclusively expressed on select tissue macrophages, in the tumor microenvironment (TME) and explores its potential as a therapeutic target in cancer. The authors present comprehensive in vitro, ex vivo, and in vivo data indicating that inhibition of VSIG4 via a monoclonal antibody (12A12) or siRNA can repolarize tumor-associated macrophages (TAMs) to a pro-inflammatory phenotype, leading to enhanced anti-tumor immune responses. The study is well-structured, presenting a logical flow from the rationale through to the experimental design and findings. The manuscript was significantly improved compared to its previous version. However, there are still some issues that definitely require clarification or correction.

Major comments

1.     Since VSIG4 expression is documented in Kupffer cells and Langerhans cells, one might anticipate effects of 12A12c administration on the liver or skin. Were any effects on these organs observed in the animal experiments? Potential side effects must be discussed in the discussion section. 

2.     What was the source of PBMCs and tumor samples? Is there ethics committee approval for the usage of human material? This must be clarified.

 Minor comments

1.     The authors use pembrolizumab term throughout the manuscript now, however in the supplementary material commercial name Keytruda is still used. Please correct to pembrolizumab also in the supplementary materials. 

Author Response

Major comments

  1. Since VSIG4 expression is documented in Kupffer cells and Langerhans cells, one might anticipate effects of 12A12c administration on the liver or skin. Were any effects on these organs observed in the animal experiments? Potential side effects must be discussed in the discussion section. 

    While we did not have a chance to perform extensive toxicology testing, we have the cage side observations data in multiple tumor models. No skin inflammation, weight loss or any other observations have been recorded in animals treated with anti-VSIG4 antibody.

    Additionally, while VSIG4 is indeed expressed on some tissue-associated macrophages, strong expression on Kupffer cells is more specific to mouse biology.  In fact, liver and skin expression is among the lowest across healthy tissues in humans (Supplementary Figure 3A). It has been added to the Discussion.

  2. What was the source of PBMCs and tumor samples? Is there ethics committee approval for the usage of human material? This must be clarified.

    Human PBMCs samples or whole blood Leukopaks were purchased from Research Blood Components, LLC. Research Blood Components follows American Association of Blood Banks guidelines for drawing donors. Donor population consisted of healthy males and females between the ages of 18 and 65. All donors completed a uniform blood donor history questionnaire. An IRB approved consent form was obtained from each donor giving permission to collect their blood and use or sell it at Research Blood Components discretion, for research purposes. Confidentiality and donor identification was assured. For more information, please see https://www.researchbloodcomponents.com/about-us

    We have added this information into Materials and Methods section.

 Minor comments

  1. The authors use pembrolizumab term throughout the manuscript now, however in the supplementary material commercial name Keytruda is still used. Please correct to pembrolizumab also in the supplementary materials. 

    We appreciate the reviewer pointing out these inconsistencies – all instances of the use ‘Keytruda’ have been corrected, now the antibody is referred to as pembrolizumab throughout the manuscript.

Reviewer 2 Report (Previous Reviewer 1)

Comments and Suggestions for Authors

It's a fascinating study. The author showed in situ hybridization of VISG4 in macrophage, but the naive readers would like to know how the antibody to identify these cell population is not available. Give the readers on the information about availability of straightforward antibody to human VSIG4, applicable to iHC in human tumor tissues mixed with TAM. OR are any efforts to obtain these antibodies in this field  known?

Author Response

It's a fascinating study. The author showed in situ hybridization of VISG4 in macrophage, but the naive readers would like to know how the antibody to identify these cell population is not available. Give the readers on the information about availability of straightforward antibody to human VSIG4, applicable to iHC in human tumor tissues mixed with TAM. OR are any efforts to obtain these antibodies in this field  known?

Thank you for drawing our attention to the antibody and probe source. For the ISH images the catalog number of the probe is listed in the figure legend of the corresponding image (Supplementary Figure 3). IHC staining of the tumor microarrays was performed with anti-VSIG4 rabbit polyclonal antibody from Abcam (Catalog # ab246869). This information was provided with the images as well. 

This manuscript is a resubmission of an earlier submission. The following is a list of the peer review reports and author responses from that submission.

Round 1

Reviewer 1 Report

Comments and Suggestions for Authors

The data are massing. The authors is a company who developed antibodies to VSIG4, which 

they intent on applying to cancer therapeutics.

1. VSIGs are a still un-famous families, then a brief introduction as to homology and structures of VSIG1 to 4. Especially when the authors argue about the specificity to human and mouse Vsig4, the readers wonder how about other VSIGs?

2. The readers like to know the existence of VSIG4 in the macrophage. The authors should show real IHC data on VSIG4 in macrophage intermingled in tumor cells.  

Reviewer 2 Report

Comments and Suggestions for Authors

The manuscript “Antibodies Targeting Human or Mouse VSIG4 Repolarize Tu-2 mor-associated Macrophages Providing the Potential of Potent 3 and Specific Clinical Anti-tumor Response Induced Across 4 Multiple Cancer Types” investigates the role of the V-set and immunoglobulin domain-containing 4 (VSIG4) protein, a B7 family-related protein exclusively expressed on select tissue macrophages, in the tumor microenvironment (TME) and explores its potential as a therapeutic target in cancer. The authors present comprehensive in vitro, ex vivo, and in vivo data indicating that inhibition of VSIG4 via a monoclonal antibody (12A12) or siRNA can repolarize tumor-associated macrophages (TAMs) to a pro-inflammatory phenotype, leading to enhanced anti-tumor immune responses. The study is well-structured, presenting a logical flow from the rationale through to the experimental design and findings. However, there are areas where the manuscript could be improved for clarity, rigor, and potential impact.

Major comments.

1.     Confirmation of the presence of VSIG4 on macrophages is presented in Figure 1; however, it is not entirely convincing. Panel B shows a representative from 2 analyzed samples. The authors need to analyze more samples and present the analysis of all of them. Panel D displays a representative result out of more than 15 experiments. The rationale for this type of presentation is unclear. The authors should present a collective analysis of all 15 experiments. It would be interesting to observe the differences between various donors.

2.     Analysis of VSIG4 in clinical samples in this manuscript is limited to mRNA expression, which is likely one of the most significant shortcomings of the study. The authors should analyze tissue and cell expression of the VSIG4 protein using immunohistochemistry or immunofluorescence analysis. This information is crucial to determine whether protein expression is confined to TAMs and to understand the intracellular localization of the VSIG4 protein.

3.     In Section 2.3 of the results, crucial controls are absent. For instance, the phenotype transition from M2c to M1 is facilitated by treating cells with an antibody or isotype, followed by LPS treatment. It is unclear what the outcome would be for M2c cells treated solely with LPS, or the effect of the antibody without subsequent LPS treatment. Furthermore, the basis for selecting specific cytokines for analysis has not been elaborated. Given the discussion about M1, including TNF and IL1beta in the analysis would seem logical.

4.     The analysis of 12A12c effect on M2 macrophages is confined to the expression of CD163. In my opinion, the authors should broaden the range of M2 markers under investigation to include, for example, the analysis of CCL18, IL1RA, and some scavenger receptors.

5.     The binding of 12A12c to VSIG4 is only studied on CHO-K1 cells overexpressing the molecule. The authors must conduct binding studies on monocyte-derived human macrophages, including M1, M2, M2c, as well as on macrophages with knocked-down VSIG4 expression.

6.     Since VSIG4 expression is documented in Kupffer cells and Langerhans cells, one might anticipate effects of 12A12c administration on the liver or skin. Were any effects on these organs observed in the animal experiments? Potential side effects have to be discussed in the discussion section. 

7.     What was the source of PBMCs? Is there ethics committee approval for the usage of human material?

8.     The authors used 10%FBS for cultivating their macrophages. This is not really the state-of-the-art way of cultivating macrophages, especially now when so many various types of serum free culture medium are available on the market. FBS is known to contain high concentrations of GM-CSF that can activate human macrophages.

 Minor issues.

1.     The authors used pembrolizumab as a control antibody in several experiments. This name should be consistently used throughout the document, avoiding the use of its commercial name, Keytruda.

2.     Text on some figures is too small to read (Figure 1A, Fig 5A,B)